# Analysis of Erosion Characteristics and Erosion Mechanism of Polypropylene Fiber Tailings Recycled Concrete in Salt Spray Environment

**DOI:** 10.3390/polym14235137

**Published:** 2022-11-25

**Authors:** Xiuyun Chen, Tao Li, Meng Zhan, Yijie Ding, Faguang Leng, Jia Sun

**Affiliations:** 1College of Architecture Engineering, Huanghuai University, Zhumadian 463000, China; 2School of Civil Engineering, Xi’an University of Architecture and Technology, Xi’an 710055, China; 3China Academy of Building Research, Beijing 100013, China

**Keywords:** salt spray environment, polypropylene fiber, mechanical properties, erosion characteristics, corrosion mechanism

## Abstract

Economic development and infrastructure improvement will inevitably lead to the accumulation of construction waste and tailings, which has not only a huge impact on the environment but is also a waste of resources. Recycling these resources and making green concrete is an effective way to solve these problems. In this study, the salt spray erosion characteristics and erosion mechanism of tailings recycled concrete (TRC) with polypropylene fibers were studied through macro and micro methods. The results showed that its compressive strength and splitting tensile strength increased at first and then decreased, with the optimum content of 0.6–0.9%, and the strength increase coefficient reached its maximum value at the erosion period being 14 d to 28 d. Under the same erosion cycle, when the fiber content was low (≤0.6–0.9%), the erosion depth hardly fluctuated. While the fiber content changed from 0.6% to 1.2%, the erosion depth and curing ability (erosion for 90 days) increased by 16.29% and 11.20%, which implied that its erosion resistance decreased sharply. Through SEM microscopic analysis, it could be observed that when the fiber content was low, the matrix structure and porosity had little change; while the fiber content was excessive, the porosity increased greatly. The longer the erosion period was, the greater the cumulative expansion of salt crystals was, and the larger the porosity was, whose results were in good agreement with the experimental results. This research provides a significant theoretical basis for the application of TRC in engineering.

## 1. Introduction

Rapid economic development has brought great convenience to our life, but it has also led to the rapid accumulation of construction waste and industrial by-products, which has posed a huge threat to our living environment and is also contrary to the goal of “carbon emissions peak and carbon neutrality”. In China, there is a great number of reserves of abandoned construction waste and iron tailings, but the utilization rate is only 30%, which is far lower than the goal of 90% for developed countries [1]. The main reason is that neither of these materials is a qualified standard building material; research has shown that they all had defects to varying degrees. TRC, which is made of these two materials, needs to be mixed with other performance-enhancing materials to change one aspect of its performance, and the optimum incorporation ratio needs to be determined.

Fiber is the main admixture material used to improve the performance of concrete. Anthony [2] studied the bending and impact properties of concrete with different abaca fiber content by means of macro and micro methods, and the results showed that when the fiber content was 0.5%, its bending strength and impact strength reached their optimum. Kumar [3] investigated the influence of different length diameter ratios and different proportions of mixed fiber on the structural performance of self-compacting concrete, and the results indicated that the combination of glass and steel fiber has better tensile strength and bending strength. Curosu [4] modified polyvinyl alcohol fiber using a chemical method and studied the crack-bridging behavior of its cement-based composite. Gonzalez [5] analyzed the flexural performance of fiber-reinforced recycled concrete, and the results showed that the addition of fiber could greatly improve the flexural performance of recycled concrete materials. Ahmad [6] conducted an experimental study regarding the strength and durability of concrete with different levels of nylon fiber content. The results indicated that when the nylon fiber content was 1.5%, its parameters of mechanical properties and durability reached their optimum. Kariminpour [7] researched the flexural performance of recycled concrete beams with different steel fiber and polypropylene fiber contents and proved that the improvement effect of polypropylene fiber on the flexural capacity of beams was higher than that of steel fiber. Sun [8] studied the influence of fiber type and steel fiber content on the impact resistance of recycled concrete, indicating that with the increase in steel fiber content, the impact resistance of recycled concrete specimens was gradually improved. Ke [9] systematically summarized the development status of recycled concrete with waste fibers and specifically analyzed the influence of different waste fibers on the mechanical properties and durability of recycled concrete.

The results show that fiber can greatly improve the crack resistance and deformation ability of cement-based composites. However, in the specific service process, it will suffer from the erosion of various external environments. Different erosion environments produce different erosion characteristics. Kawamorita [10] evaluated the frost damage resistance of concrete using low-heat portland cement and fly ash, which was considered to have sufficient practicality. Manjunath [11] developed a new high-performance self-compacting alkali slag concrete using three industrial by-products from the steel industry, evaluated its strength performance, and discussed the durability of this kind of mixture in long-term exposure to acid, sulfate, chloride, and other corrosive environments. Zhu [12] studied the high-temperature resistance of TRC under a high-temperature environment. The results indicated that the recycled concrete column containing iron tailings sand displayed better resistance after exposure to fire. Huang [13] analyzed the carbonization and macro mechanical properties of TRC and proved that when the iron tailings content was 30%, the performance of recycled concrete reached the level of ordinary concrete.

Concrete with the same components will show different performances in different environments. China has a vast coastline and a large area of inland salt lakes, and concrete structures in this region are exposed to corrosive salt spray environments. Therefore, it was necessary to study the erosion characteristics and erosion mechanism of concrete in this type of environment. Chendra [14] analyzed the main factors that affect the adsorption capacity of chloride ions on the surface of concrete under a salt spray environment via a wind tunnel test. Ariyachandra [15] analyzed the chloride diffusion capacity and binding capacity of recycled concrete and then proposed the chloride-erosion-resistance mechanism of recycled concrete. Mohamed [16] developed a new artificial neural network calculation model for the prediction of the chloride permeability of self-compaction concrete, which was in good agreement with the test results. Professor Wang Sheliang from Xi’an University of Architecture and Technology conducted a series of detailed studies regarding the carbonization [17,18], salt spray erosion [19], freeze–thaw [20], and sulfate dry–wet cycle [21] of TRC, making great contributions to the popularization and application of recycled concrete.

To reduce the adverse impact of waste resources on the environment and make full use of the excellent splicing effect of fibers, a new type of green concrete was made, and its erosion performance in complex environments was studied. Therefore, in this study, considering different content levels of waste fiber, the corrosion resistance characteristics, and the corrosion mechanism of TRC in a salt spray environment were studied.

## 2. Test Setup

### 2.1. Test Materials

The physical and mechanical properties of conventional materials such as coarse and fine aggregates and materials such as iron tailings and polypropylene fiber used in the test have previously been discussed in the literature [17], and they met the Chinese specification requirements [22,23], which are not repeated here.

### 2.2. Test Mix Proportion

In accordance with the previous research results relating to the coarse aggregates used [17,21,24], the amount of recycled aggregate and iron tailings in the mixture was set to 30%, and the mix proportion was designed according to the previous literature [25,26]. The water–cement ratio and sand ratio were set to 0.4 and 0.35, respectively. The polypropylene fiber content was set to 0%, 0.3%, 0.6%, 0.9%, and 1.2%. Natural aggregate concrete (NAC) and recycled aggregate concrete (RAC) were used as reference blocks. After a trial mixture was made and adjustment was carried out, the proportions of the mixtures of concrete under various working conditions were determined and are shown in Table 1.

### 2.3. Test Parameter Setting

To facilitate the rapid salt spray erosion test, a cube test block (100 mm × 100 mm × 100 mm) was used in the test. As it eliminated the adverse effects of multi-dimensional salt spray erosion on erosion depth and ion concentration, Vaseline was used to seal five surfaces, leaving one non-poured surface as the only erosion surface.

The test was mainly composed of five steps, and the parameter settings of each step are shown in Table 2.

During the test, we used a large salt spray corrosion test box produced by Wuxi Yingbai Technology Co., Ltd. (Wuxi, China), as shown in Figure 1. When the erosion had taken place for the specified period (7 d, 14 d, 28 d, and 90 d), the cube test block was taken out and wiped clean, and the compressive strength, splitting tensile strength, erosion depth, and erosion ion concentration were determined. The process of determining erosion depth and erosion ion concentration was as follows:

(1) Erosion depth measurement: The block was split along the erosion surface, 0.1 mol/L AgNO_3_ solution was sprayed, and after 15 min, 10 points on the cleavage surface were measured with a digital depth meter, and the average value was determined.

(2) Ion concentration value: The single-side grinding method was used to obtain powder from the eroded surface every 2 mm, layer by layer, to 10 mm, and then, holes were drilled every 5 mm to obtain a powder. Finally, the powder was screened out through a 0.16 mm sieve. Then, according to the specification [27], the total chloride ion concentration, *C_t_*, and free chloride ion concentration, *C_f_*, were extracted with dilute nitric acid and distilled water and titrated with potassium thiocyanate and potassium chromate, respectively. The solid–liquid extraction method [28] was used to determine the mass fraction.

## 3. Results and Discussion

### 3.1. Cube Compressive Strength Value

Figure 2 indicates the influence of different waste fiber content on cube compressive strength and its coefficient of increase. As can be seen from Figure 2a, with the increase in fiber content, there was an approximate trend of first increasing and then decreasing at the same erosion age, but the increase was relatively limited. The peak point was approximately 0.6~0.9%, and the intensity value increased by 4.73% (0%), 4.09% (0.3%), 4.07% (0.6%), 6.43% (0.9%), and 11.42% (1.2%), respectively. At this moment, the optimal mixing ratios given in the literature [29,30] were 1% and 0.2%. This is mainly due to the limited effect of fiber bonding on compressive strength. The pull effect of fiber was enhanced, and the drop rate after the peak point was relatively slow with the increase in fiber content. When the content was high (1.2%), the porosity increased greatly due to the fiber space network structure. Therefore, when the corrosion lasted for 90 days, the filling effect and expansion densification effect of salt spray erosion products on the pore structure [31] made the porosity of the test block smaller, and the strength value had a local growth effect after the peak point.

Figure 2b demonstrates the change in the strength growth coefficient in different erosion cycles under the same waste fiber content. It can be seen that the peak point of RAC-21 and PE-RAC-5 (1.2%) appeared on the 28th day of erosion, while the rest appeared on the 14th day of erosion, but the change was relatively gentle. Due to the large porosity of RAC-21, the crystallization and filling effects of erosion were obvious, and the increase coefficient of its peak point was the largest, reaching 1.16. At the same time, the recycled concrete had increased density due to the addition of 30% tailings. In addition, the tensile transition effect of fibers made the erosion age more sensitive. For NAC-21 and PE-RAC-3 (0.6%), deterioration occurred after 28 days of erosion. Therefore, the amount of waste fiber should be strictly controlled during use.

### 3.2. Splitting Tensile Strength

Figure 3 displays the change rule of splitting the tensile strength of waste fiber content at different erosion ages. Compared to the good tensile effect and toughening effect of waste fiber, its effect on tensile strength was more significant. It can be seen from Figure 2a that the tensile strength was at its maximum with the mixing amount of 0.6~0.9% at each erosion age, which was higher than the mixing amounts of 0.1% and 0.12% reported in the literature [29,32], but the same change rule existed. When its content changed from 0% to 0.9%, the strength also increased by 83.8% (0 d), 71.6% (7 d), 45.3% (14 d), 80.5% (28 d), and 89.9% (90 d), respectively; this effect was quite obvious. In Figure 2b, except for RAC-21 and PE-RAC-1, the increase coefficients are all below 10%, indicating that the increase ratio of tensile strength caused by salt spray erosion was relatively small. Similar to the impact of tailings, the peak point of concrete with large porosity occurred after 28 d of erosion. Due to the soft texture of waste fiber, the pore pressure caused by salt crystallization could be relieved and transferred, meaning the effect of erosion age on the waste fiber was relatively slight, which is visually shown by the curve after the peak point in Figure 2b, which was close to a straight line.

### 3.3. Erosion Depth

The erosion depths of salt spray under different erosion cycles are shown in Figure 4 and Figure 5. It can be seen that the longer the salt fog erosion took place, the greater the erosion depth was. With the increase in fiber content, the erosion depth decreased first and then increased, but the increasing trend was different. When the fiber content was 0.6%, the erosion depth reached its minimum. This phenomenon was similar to that observed in the literature [33]. Taking 90 d of erosion as an example, the depth increased by 1.53%, while the content increased from 0% to 6%. The main reason for this was that the fiber displayed two effects. On the one hand, it had an obvious pulling effect, which meant the concrete could resist the expansion force generated by some salt crystallization. On the other hand, it increased the contact area between different materials, which would lead to an increase in porosity to some extent. When the dosage was small (≤0.6%), it had little effect on the erosion depth. However, when the dosage changed from 0.6% to 1.2%, the erosion depth increased by 16.29%, indicating that when its dosage was large, the increasing effect on the porosity of concrete was also greater.

### 3.4. Relationship between Free Chloride Content and Salt Spray Erosion Depth

In the test, the salt spray erosion concentration outside the concrete remained constant, the chloride concentration inside the concrete gradually increased with time, and it also decreased with the increase in the distance from the concrete surface. Figure 6 displays the distribution of free chloride concentration at each age under the action of waste fiber.

As can be seen, the free chloride concentration of different dosages of waste fiber presented a rapid downward trend from the surface to inside the concrete. As fiber had no active effect, it also increased the porosity of concrete to some extent, but its unique pull effect alleviated the expansion force generated by salt crystallization, and both effects worked at the same time. For example, at a depth of 2 mm after 28 days of erosion, the fiber content increased from 0% to 1.2%. Compared with RAC, the free chloride concentration changed, respectively: −14.29%, −12.3%, −15.87%, −15.08%, and 1.98%. Meanwhile, when the fiber content was lower (≤0.9%), the ability of TRC to resist chloride ion corrosion increased, but the amplitude was relatively unchanged, and the ability to alleviate salt crystal expansion became dominant. However, while the content of waste fiber was high, the increasing effect of the TRC porosity caused by waste fiber gradually became prominent, which meant its resistance to chloride ion corrosion gradually declined. According to the figure displayed above, when the fiber content reached 1.2%, its chloride ion erosion resistance was closest to that of recycled concrete. It can also be predicted that if the dosage continued to increase, its resistance would continue to decline.

### 3.5. Time-Varying Characteristics of Chloride Ion Content

In the process of salt spray erosion, free chloride ions gradually accumulate at the same depth from the concrete surface, and the content also gradually increases. In this paper, the erosion depth of 2 mm and 4 mm at each age was compared and analyzed so as to determine the trend of its temporal variation characteristics. Figure 7 exhibited the concentration distribution of free chloride ions (*C_f_*), combined chloride ions (*C_b_*), and total chloride ions (*C_t_*) at each age under different levels of waste fiber content.

Overall, the curves in the figure displayed above can be divided into two categories according to the value: one was NAC and the concrete with 1.2% fiber content (PE-RAC-5), and the rest were divided into the other category. The porosity of recycled concrete was improved due to the addition of tailings, meaning the chloride ion content of TRC (PE-RAC-1) with 0% fiber content was lower than that of RAC. The addition of appropriate fiber did not change the matrix structure of TRC too much. Therefore, the TRC with 0.6% fiber content (PE-RAC-3) differed little from PE-RAC-1 and NAC. To be specific, the addition of fiber did not affect the alkalinity of cementitious materials. The higher the fiber content was, the larger the contact area between the fiber and the concrete materials was, and the greater the porosity was. Therefore, the free chloride ion (*C_f_*) and total chloride ion (*C_t_*) contents of PE-RAC-3 were slightly higher than those of PE-RAC-1. Due to the spatial network distribution of waste fibers in concrete and the accumulation of fibers in different directions, the porosity increased greatly when waste fibers were overdoped (PE-RAC-5). Therefore, the free chloride (*C_f_*) and total chloride (*C_t_*) contents of PE-RAC-5 were equal to or even higher than those of RAC. At the same time, the greater the distance from the concrete surface, the later the position started to consume C_3_A, meaning the time when *C_f_* and *C_b_* were equal appeared later; these results are similar to those of Niu [34].

### 3.6. Chloride Ion Curing Property

The absorption and solidification of chloride ions via concrete can greatly reduce the concentration of free chloride ions and total chloride ions in concrete, as well as the transmission rate of chloride ions, thus reducing the risk of reinforcement corrosion [35]. In this paper, the complete linear model was used to determine the correlation between the free chloride ion content and the total chloride ion content in the case, namely:$$C_{t}=K_{1}\cdot C_{f}+K_{2}$$
where A is the binding coefficient and b is the adjusted constant. This formula was used to study the correlation between free chloride ion and total chloride ion content at 28 d and 90 d.

Figure 8 shows the relationship between free chloride ion and total chloride ion concentrations under different levels of waste fiber content. Due to the “wick effect” of concrete [36], the concentration of chloride ions gradually decreased with the penetration depth and slowed down when it dropped to a certain depth. Under the action with the concentration gradient of concrete, it gradually diffused into the concrete, but as it was limited by the diffusion resistance of particles and the friction resistance of pores, the diffusion rate of concrete was low [34]. Therefore, stepped numerical points appear in Figure 8a,c. With the change in aggregate type and fiber content, the pore structure and porosity of concrete also varied, which resulted in the change in free chloride ion and total chloride ion content in the same erosion cycle. The addition of fiber can effectively alleviate the expansion pressure caused by pore salt crystallization and delay the development of micro-cracks inside the specimen. The larger pores in the surface layer provided space for chloride adsorption, the internal micro-cracks supplied channels for chloride transport, and the specific change rules were also different under diverse dosages. However, different erosion ages were influenced by factors such as porosity, diffusion time, and salt crystallization pressure [37,38] and changed the chloride concentration, which led to the changes shown in Figure 8b,d.

Figure 9 demonstrates the comparison of curing coefficients with different levels of fiber content. It can be seen that when the fiber content was low (≤0.6%), there was no significant difference in the curing coefficients of different levels of fiber content due to the activity of tailings and the increase in porosity caused by fiber. However, when the amount of waste fiber was large, the space grid effect of waste fiber became significant, which made the pore structure in TRC coarser; the gross pores increased greatly [39,40]; the specific surface area became larger, and the physical adsorption of chloride ions was greater. The different levels of fiber content had no influence on the alkali composition of concrete due to the absence of additional active materials. In general, when the content of waste fiber was high, the curing effect on chloride ions became stronger. This was also the main reason why the chloride ion curing capacity of 28 d and 90 d corroded concrete changed by −5.19%, −5.14%, 4.82%, and 11.20%, respectively, compared with NAC when the fiber content was increased from 0.6% (PE-RAC-3) to 1.2% (PE-RAC-5).

### 3.7. Salt Spray Erosion Mechanism

Scanning electron microscopy (SEM) was used to conduct SEM analysis on the test blocks under different salt spray erosion cycles, as shown in Figure 10.

As can be seen from this figure, when salt spray erosion took place for 28 d and 90 d, clusters of crystalline products could be observed. Under the same working conditions, the longer the erosion time was, the more crystallized the products were and the more obvious the cluster-like products were. The addition of recycled aggregate made the matrix structure of the test block relatively loose and also provided space for the generation, growth, and development of chloride crystals. It is intuitively shown in the figures above that the matrix structure of concrete was covered by salt crystals due to the generation of salt crystals and the existence of physical expansion pressure of salt crystals, and at the same time, some connected cracks were gradually transformed into closed pores. The harmful hole was further transformed into a harmless hole.

When the erosion age was longer (90 d), it can be clearly seen that there was a significantly larger number of pores and micro-cracks in the concrete matrix structure than those in the working condition with shorter erosion ages (28 d), as shown in Figure 10b,d,f,h,j, which shows that when the erosion time was longer, the matrix structure of concrete was also damaged with the accumulation of salt crystals. The reason for this was that, on the one hand, the physical expansion of chloride crystals that accumulated in the pores made the matrix structure produce tensile stress and made it easy to crack; on the other hand, there was relatively little free water in the concrete, and the generation of cement hydration and chloride crystals needed to consume free water, resulting in adverse effects on the matrix structure due to water migration. In conclusion, the mechanical properties and durability of almost all working conditions decreased by varying degrees when the erosion age was longer. Xue [41] and Yue [42] reached similar conclusions in their research.

Figure 10f,h,j represented the cases in which the fiber contents were 0%, 0.6%, and 1.2%. As 30% tailings were added, the porosity of RAC was effectively improved, and there were not many harmful cracks and holes in the matrix structure, which suggests that there was little difference in the overall compressive performance and durability of recycled concrete. However, with the increase in the amount of fiber, the contact area of different materials in the concrete increased, which led to a slight increase in porosity. Therefore, the compressive strength increased slightly, and the erosion depth did not change too much, with the tensile strength of the fiber was greatly increased due to its unique pull joint effect. Due to the increase in the amount of fiber, the increase in porosity caused by the contact surface of two different materials could not be ignored. Figure 10j shows that there were no harmful cracks, but the number of harmful holes and pores increased significantly, which also led to a decrease in the compressive property, caused little change to the tensile property, and caused a significant increase in the erosion depth when the fiber content was high.

Based on the above analysis, salt spray erosion was a complex physical and chemical reaction process. The chloride ions that migrated into the concrete were absorbed by cement hydrates through physical adsorption, which was relatively weak and easy to be destroyed, and then would be converted into free chloride ions after destruction. The other was to react with cement hydrate through a chemical bond, which remained relatively stable. The common Friedel salt was this kind of combination. The chemical equation was as follows:(1)$$3\mathrm{CaO}\cdot {\mathrm{Al}}_{2}O_{3}\cdot 6H_{2}O+{\mathrm{Ca}}^{2+}+2{\mathrm{Cl}}^{-}+4H_{2}O\to 3\mathrm{CaO}\cdot {\mathrm{Al}}_{2}O_{3}\cdot {\mathrm{CaCl}}_{2}\cdot 10H_{2}O$$

The remaining part was free chloride ions dissolved in the pore solution in the concrete, which can continue to migrate to the interior of the concrete under the driving of the concentration difference. When it migrated to the surface of the reinforcement, it would destroy the passive film formed by the reinforcement in an alkaline solution. When the free chloride ion accumulated to a certain concentration on the surface of the reinforcement, it would greatly reduce the pH value of the reinforcement surface, causing corrosion and expansion. Therefore, to avoid the corrosion of the reinforcement, the content of free chloride ions should be strictly noted.

In conclusion, waste fiber can improve the splitting tensile strength of green concrete, but it also makes the contact area of different materials in concrete increase and affects its corrosion resistance. However, with careful consideration, when using fibers in a salt spray environment, it was necessary to strictly control its incorporation amount to be under 0.6% in order to obtain the best erosion resistance.

## 4. Conclusions

To accelerate the recovery and utilization of recycled aggregate and tailings, the salt spray erosion characteristics and erosion mechanism of TRC with different waste fibers under the salt spray erosion environment were analyzed, and the following conclusions could be drawn:

(1) At the same erosion age, the compressive strength and splitting tensile strength increased first and then decreased with the increase in fiber content, and the peak point was 0.6–0.9%. Meanwhile, each strength index first increased and then decreased with the increase in the salt spray erosion time. When the erosion time was 14 d to 28 d, its peak coefficient reached the maximum value.

(2) In the same erosion cycle, when the fiber content was low (≤0.6%), the erosion depth changed little. As the fiber content increased from 0.6% to 1.2%, the erosion depth increased by 16.29%, causing the erosion resistance to be worse.

(3) The free chloride ions under different levels of fiber content showed a sharp decline from outside to inside the concrete. When the content of fiber was low (≤0.9%), the chloride resistance of TRC was enhanced, but the increased amplitude was almost the same. When the dosage was high, the anti-chloride ion erosion ability decreased further, which indicated that the free chloride ion content in the same erosion cycle and location became high.

(4) When the fiber content was low (≤0.6%), the curing coefficients of different fibers changed little. However, the higher the fiber content, the stronger the curing effect of the fiber on chloride ions was. When the fiber content increased from 0.6% to 1.2%, the curing capacity increased by 11.20% after 90 days of erosion.

(5) Through the SEM microscopic analysis of different erosion cycles, it was shown that the addition of a small amount of fiber had little effect on the matrix structure and porosity of TRC, but when the content was high, the porosity was greatly increased. In the process of salt spray erosion, due to the formation of salt-bound crystals and the existence of salt-bound crystal expansion pressure, when the fiber content was low, the mechanical properties and resistance to salt spray erosion of TRC were slightly increased. On the contrary, due to the cumulative expansion of salt-bound crystals, its salt spray resistance deteriorated more when the fiber content was high.

## Figures and Tables

**Figure 1 polymers-14-05137-f001:**
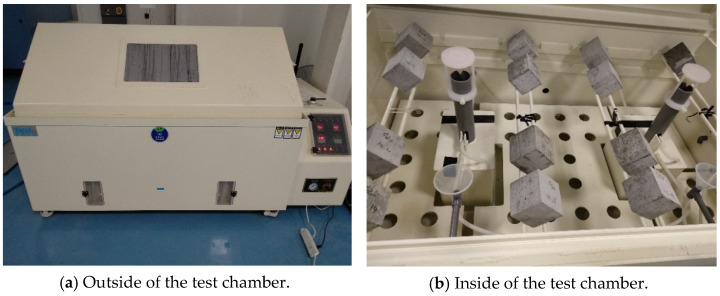
Salt spray corrosion test chamber.

**Figure 2 polymers-14-05137-f002:**
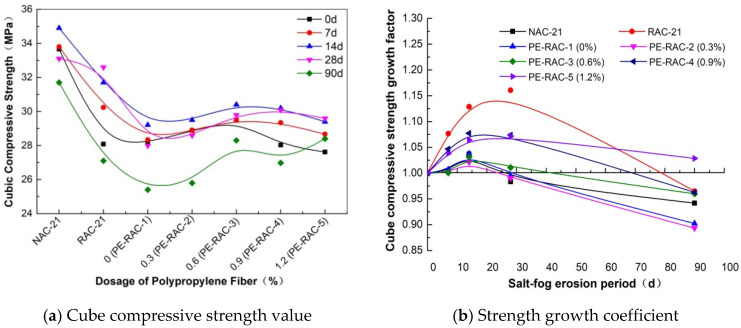
Influence of waste fiber content on cube compressive strength.

**Figure 3 polymers-14-05137-f003:**
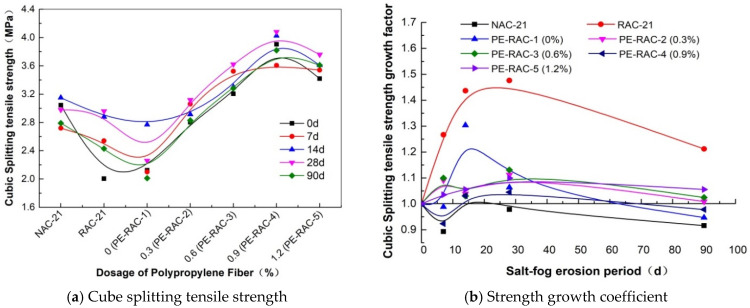
Influence of waste fiber content on splitting tensile strength of cube.

**Figure 4 polymers-14-05137-f004:**
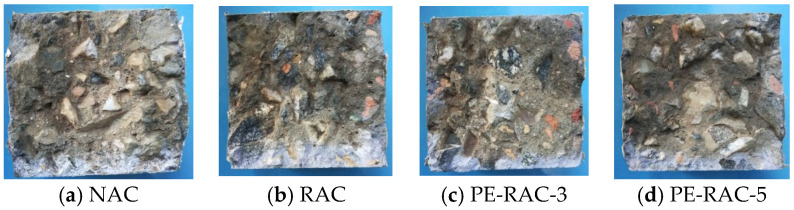
Erosion map of salt spray with different fiber content (14 d).

**Figure 5 polymers-14-05137-f005:**
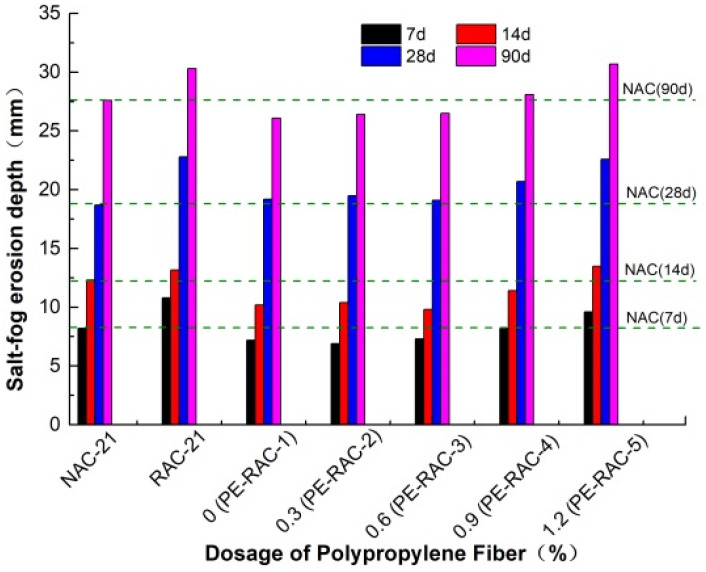
Salt spray erosion depth under different waste fiber content.

**Figure 6 polymers-14-05137-f006:**
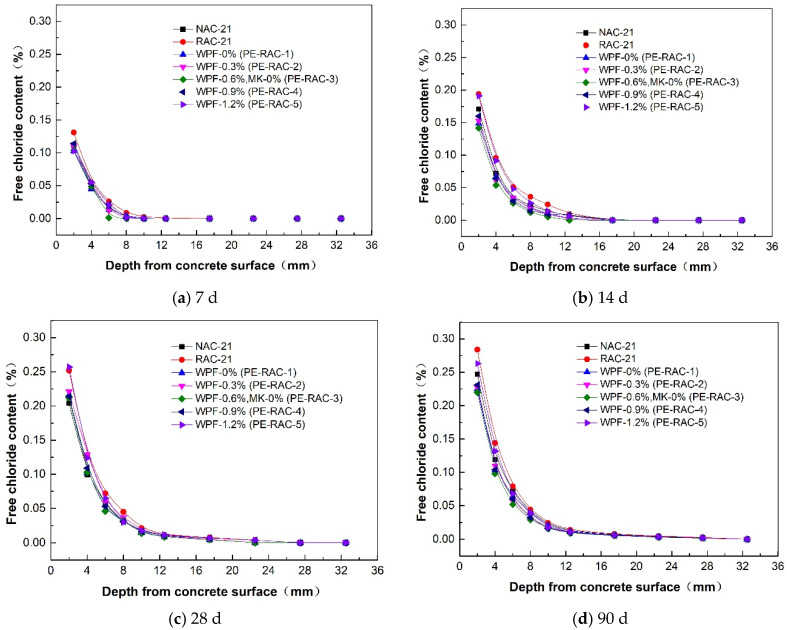
Relationship between free chloride content and salt spray erosion depth.

**Figure 7 polymers-14-05137-f007:**
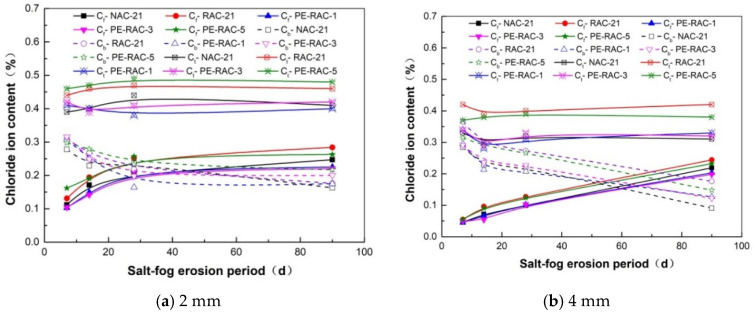
Distribution of ion concentration related to each age (*C_f_*, *C_b_*, and *C_t_*).

**Figure 8 polymers-14-05137-f008:**
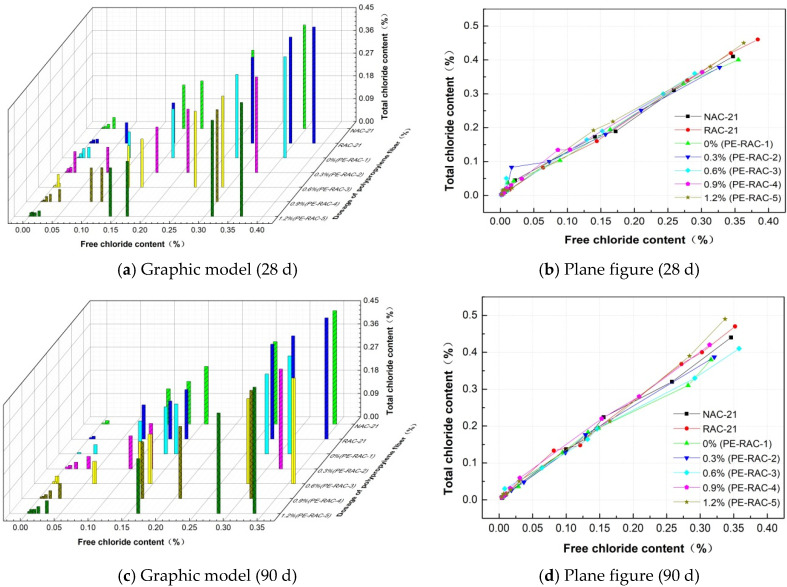
Relationship between free chloride ion concentration and total chloride ion concentration.

**Figure 9 polymers-14-05137-f009:**
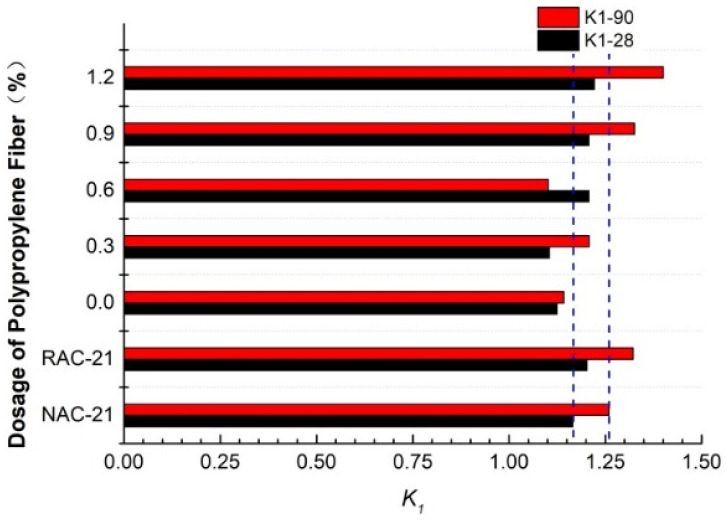
Comparative analysis diagram of curing coefficient in 28 d and 90 d.

**Figure 10 polymers-14-05137-f010:**
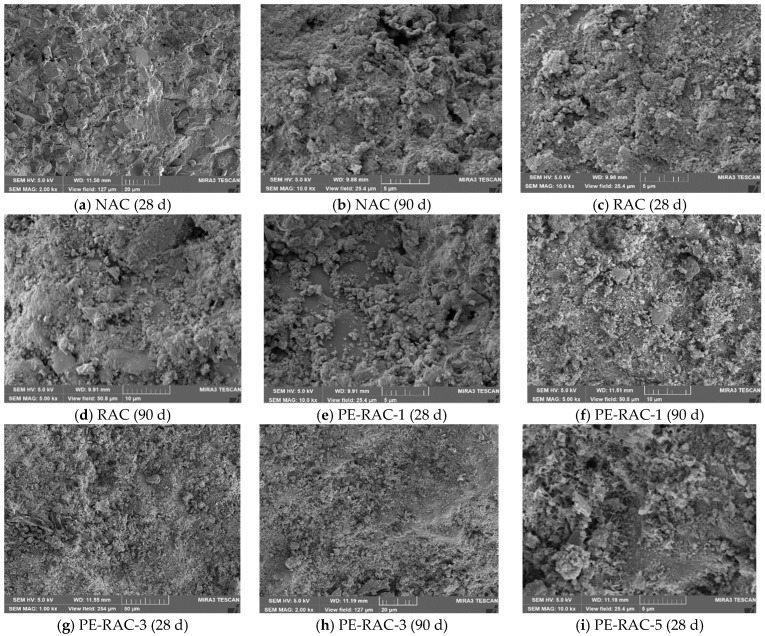
SEM scanning micromorphology of salt spray erosion in different cycles.

**Table 1 polymers-14-05137-t001:** Proportions of mixtures under different working conditions (kg/m^3^).

Test Block Number	Cementitious Material	Coarse Aggregate		Fine Aggregate		Water	Fibers	Water Reducer
		NCA	RCA	Sand	IOT			
NAC-21	538	1063	0	572	0	215	0	0
RAC-21	538	735	315	566	0	215	0	0
PE-RAC-1	538	748	320	403	173	215	0	8.07
PE-RAC-2	538	748	320	403	173	215	1.614	8.07
PE-RAC-3	538	748	320	403	172	215	3.228	8.07
PE-RAC-4	538	748	320	403	172	215	4.842	8.07
PE-RAC-5	538	748	320	403	172	215	6.416	8.07

**Table 2 polymers-14-05137-t002:** Main process of salt spray erosion test.

Main Steps	1. Making and Curing	2. Drying	3. Covering	4. Salt Spray Erosion	5. Testing
Key parameters setting	Standard curing for 28 d	Drying for 48 h at 60 °C	Covering five surfaces with paraffin and leaving an erosion surface (non-pouring surface)	NaCl: 5% pH: 6.5–7.2 Temperature: (35 ± 2) °C	Loading rate: 0.5 MPa/s (compressive strength); 0.05 MPa/s (Tensile strength)

## Data Availability

The data presented in this study are available on request from the corresponding author.

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
