# Peer review of "Analysis of Erosion Characteristics and Erosion Mechanism of Polypropylene Fiber Tailings Recycled Concrete in Salt Spray Environment"

_polymers, 2022, doi:10.3390/polym14235137_

Round 1

Reviewer 1 Report

The conducted research is interesting and innovative. Following changes are very important if the paper is going to be published  

1. The abstract must include the reason of conducting the particular research

2. The problem statement is not clearly defined in the introduction part

3. Test setup images are missing

4. provide detailed procedure clearly

5. More images are needed and must be very clear

6. conclusions need further revision

Author Response

Dear Editors and Reviewers:

We quite appreciate for your favorite consideration and the reviewers’ insightful comments concerning our manuscript entitled “Analysis of erosion characteristics and erosion mechanism of polypropylene fiber tailings recycled concrete in salt spray environment (Log ID: Polymers-1987459). Those comments are all valuable and helpful for revising and improving our paper, as well as the important guiding significance to our research. We have studied comments carefully and have made correction which we hope to meet with approval. We hope this revision can make our paper more acceptable. A revised manuscript with the correction sections was attached as the supplemental material and for easy check/editing purpose. Should you have any questions, please contact us without hesitate. The main corrections in the paper and the responds to the reviewer’s comments are as following:

Reviewer #1: The conducted research is interesting and innovative. Following changes are very important if the paper is going to be published. [The revised places appear in the form of yellow shadows in the article.]

(1) The abstract must include the reason of conducting the particular research. 

Response: Thank you for your careful review. Indeed, the purpose of this study can be reduced in the past. Thank you. It has been modified in the text. See the yellow shaded part of the “Abstract Section” in the text for details, as shown in line 12-15. We replaced “To improve the utilization efficiency of construction waste, waste polypropylene fiber was selected as a strengthening agent and applied to tailings recycled concrete (TRC).” with “To reduce the impact of recycled aggregate and tailings on the environment, and accelerate the recycling efficiency of waste resources, the performance of tailings recycled concrete (TRC) was improved by waste fiber, and its salt spray erosion characteristics and erosion mechanism were studied.”

(2) The problem statement is not clearly defined in the introduction part.   

Response: Thank you. To lead to the theme of the article, the introduction part has written three pieces of content, one is the accumulation of waste materials that cannot be disposed of, the other is the rule of the influence of fiber on concrete performance, and the last piece is the influence of special environment on concrete performance. Our understanding is that our final summary is not quite in place. Therefore, the last paragraph of the introduction has been re transited in the text. See the yellow part of the introduction, as shown in line 93-96.

(3) Test setup images are missing.

Response: Thank you for your valuable advice. The picture of the salt spray erosion test device has been added in the article, as shown in Figure 1 of the article. Thank you.

(4) Provide detailed procedure clearly.

Response: Thank you for your reminder. Whether it is the salt spray erosion depth or the related chloride ion content, there are clear provisions in China's national specifications (Literature 27). We have also consulted the relevant literature at home and abroad, and the measurement methods and principles are not very different. Therefore, the paper has not carried out a detailed development. After communication with our research group, we felt that it was not necessary to carry out detailed development, so we did not add new ones for the time being. Maybe our understanding is biased. If you feel that it is necessary to add this part, we will add it later.

(5) More images are needed and must be very clear.

Response: Thank you for your valuable suggestions. The content is displayed in pictures, which is more intuitive and easier to be accepted by everyone. However, the pictures are mainly used to support the content of the article, and relevant test device diagrams are added to the article. As for the test results, each part of the test results has a clear picture to express, so we feel that we cannot add more pictures. For the clarity of the plate, there are some pictures that are not very clear, and we have made more adjustments, see the pictures in the text for details. Thank you.

(6) Conclusions need further revision.

Response: Thank you for your beneficial suggestions. The Conclusions in this article have been adjusted, as shown in line 327-355, Conclusions section. Thank you!

Special thanks to you for your good comments. We try our best to improve the manuscript and make some changes in the manuscript. Most of the changes are prompted in the comments, some grammatical or typographical errors have been revised with no mark, which changes will not influence the content and framework of the paper.

We appreciate for Editors/Reviewers’ warm work earnestly, and hope that the correction will meet with approval.

Once again, thank you very much for your comments and suggestions.

Reviewer 2 Report

The article, "Analysis of erosion characteristics and erosion mechanism of polypropylene fiber tailings recycled concrete in salt spray environment," focuses on an interesting and current issue related to the corrosion mechanism and its impact on material strength. However, some elements of the article need improvement to enhance the quality of the manuscript.

The article should be expanded to include a section - "discussion of results" and reference to other studies also from beyond China. The literature review should also be expanded. This will allow more applicability of the presented results and make a more significant contribution to the development of research related to the analysis of the corrosion mechanism.

Author Response

Dear Editors and Reviewers:

We quite appreciate for your favorite consideration and the reviewers’ insightful comments concerning our manuscript entitled “Analysis of erosion characteristics and erosion mechanism of polypropylene fiber tailings recycled concrete in salt spray environment (Log ID: Polymers-1987459). Those comments are all valuable and helpful for revising and improving our paper, as well as the important guiding significance to our research. We have studied comments carefully and have made correction which we hope to meet with approval. We hope this revision can make our paper more acceptable. A revised manuscript with the correction sections was attached as the supplemental material and for easy check/editing purpose. Should you have any questions, please contact us without hesitate. The main corrections in the paper and the responds to the reviewer’s comments are as following:

Reviewer #2:The article, "Analysis of erosion characteristics and erosion mechanism of polypropylene fiber tailings recycled concrete in salt spray environment," focuses on an interesting and current issue related to the corrosion mechanism and its impact on material strength. However, some elements of the article need improvement to enhance the quality of the manuscript.

The article should be expanded to include a section - "discussion of results" and reference to other studies also from beyond China. The literature review should also be expanded. This will allow more applicability of the presented results and make a more significant contribution to the development of research related to the analysis of the corrosion mechanism.

Response: Thank you for your beneficial suggestions. In the “discussion of results” section, we scattered the contents to the bottom of each index for discussion , and compared them with the relevant literature. For details, see the red mark in the article, and relevant literature is also cited at the end of the article. However, because there are many kinds of fiber, corrosive environment and concrete, and there is no comparability in many cases, some indicators have not found similar foreign literature, so some indicators have not participated in the comparison.

[29] Alwesabi E. A.H., Abu Bakar B.H., Alshaik, I. M.H., Akil H. Md. Experimental investigation on mechanical properties of plain and rubberised concretes with steel–polypropylene hybrid fibre [J]. Constr. Build. Mater., 2020,233:117194

[30] Zeyad A.M., Khan A.H., Tayeh B.A. Durability and strength characteristics of high-strength concrete incorporated with volcanic pumice powder and polypropylene fibers[J]. J. Mater. Res. Technol.,2019.

[32] Guo Hui, Tao Junlin, Chen Yu, et al. Effect of steel and polypropylene fibers on the quasi-static and dynamic splitting tensile properties of high-strength concrete[J]. Construct. Build. Mater.,2019,224: 504-514.

[33] Fiore, V., Sanfilippo, C., Calabrese, L. Dynamic Mechanical Behavior Analysis of Flax/Jute Fiber-Reinforced Composites under Salt-Fog Spray Environment[J]. Polymers, 2020,12:716. https://doi.org/10.3390./polym12030716.

Reviewer 3 Report

·        The language of this paper must be improved. For example, L31, what is “carbon peaking”? This is not recognized worldwide. On the same page at L32, please avoid using “amazing” and “not enough 30%” and write more precisely. The whole paper looks like being translated from software, this cannot be accepted.

·        The title numbers are wrong. The “Test setup” should be the section 2.

·        The pencils in Figure 1 look weird, it is suggested to use a regular chart.

·        Same language problem from L121 to L130, the salt-spray test description is poor.

·        Fig.2 should be presented after the following paragraph. Same as the following figures.

·        All figures in Figure 8 are not clear and must be improved.

·        The conclusions must be improved. 

Author Response

Dear Editors and Reviewers:

We quite appreciate for your favorite consideration and the reviewers’ insightful comments concerning our manuscript entitled “Analysis of erosion characteristics and erosion mechanism of polypropylene fiber tailings recycled concrete in salt spray environment (Log ID: Polymers-1987459). Those comments are all valuable and helpful for revising and improving our paper, as well as the important guiding significance to our research. We have studied comments carefully and have made correction which we hope to meet with approval. We hope this revision can make our paper more acceptable. A revised manuscript with the correction sections was attached as the supplemental material and for easy check/editing purpose. Should you have any questions, please contact us without hesitate. The main corrections in the paper and the responds to the reviewer’s comments are as following:

Reviewer #3: The language of this paper must be improved. For example, L31, what is “carbon peaking”? This is not recognized worldwide. On the same page at L32, please avoid using “amazing” and “not enough 30%” and write more precisely. The whole paper looks like being translated from software, this cannot be accepted.

Response: I apologize for our poor English. Thank you for your advice. We found a professional person to revise the grammar. I hope this time there will be no problem. Relevant modifications are indicated in the article with green background, thank you.

1. The title numbers are wrong. The “Test setup” should be the section 2.

Response: Thank you for your careful review. This is our carelessness, the section has been revised in the text.

2. The pencils in Figure 1 look weird, it is suggested to use a regular chart.

Response: Thank you for your suggestion. It has been modified in the text, and the content in the figure is reflected in the form of a table. Thank you!

Table 1. Main process of salt spray erosion test.

3. Same language problem from L121 to L130, the salt-spray test description is poor.

Response: Thank you for your reminder. Whether it is the salt spray erosion depth or the related chloride ion content, there are clear provisions in China's national specifications (Literature 27). We have also consulted the relevant literature at home and abroad, and the measurement methods and principles are not very different. Therefore, the paper has not carried out a detailed development. After communication with our research group, we felt that it was not necessary to carry out detailed development, so we did not add new ones for the time being. Maybe our understanding is biased. If you feel that it is necessary to add this part, we will add it later.

4. Fig.2 should be presented after the following paragraph. Same as the following figures.

Response: Thank you for your careful review. This was previously adjusted so that the image and the image name below can be placed on a whole page. Other similar issues in the article have also been adjusted, thank you.

5. All figures in Figure 8 are not clear and must be improved.

Response: Thank you. Figure 8 shows the three-dimensional relationship between total chloride ion content, free chloride ion content and iron tailings content. In order to save space, a three-dimensional model is adopted. At the same time, the relationship between free chloride ion content and total chloride ion content is shown on the right, which is mainly an explanation of the picture on the left. We think it can also be diluted clearly! If we really need to modify it, we can also modify it into a flat picture.

6. The conclusions must be improved.

Response: Thank you for your beneficial suggestions. The Conclusions in this article have been adjusted, as shown in line 327-355, Conclusions section. Thank you!

Special thanks to you for your good comments. We try our best to improve the manuscript and make some changes in the manuscript. Most of the changes are prompted in the comments, some grammatical or typographical errors have been revised with no mark, which changes will not influence the content and framework of the paper.

We appreciate for Editors/Reviewers’ warm work earnestly, and hope that the correction will meet with approval.

Once again, thank you very much for your comments and suggestions.